# Effects of Environmental Conditions on the Individual Architectures and Photosynthetic Performances of Three Species in *Drosera*

**DOI:** 10.3390/ijms24129823

**Published:** 2023-06-06

**Authors:** Krzysztof Banaś, Rafał Ronowski, Paweł Marciniak

**Affiliations:** Department of Plant Ecology, Faculty of Biology, University of Gdansk, Wita Stwosza 59, 80-308 Gdańsk, Poland; rafal.ronowski@ug.edu.pl (R.R.); p.marciniak.841@studms.ug.edu.pl (P.M.)

**Keywords:** *Drosera anglica*, *D. intermedia*, *D. rotundifolia*, *Drosera* habitats, substrate properties, Fv/Fm ratio, photosynthetic efficiency

## Abstract

The aim of this study was to determine the environmental conditions, individual architectures, and photosynthetic efficiencies of three sundew species: *Drosera rotundifolia*, *D. anglica*, and *D. intermedia*, found in well-preserved peatlands and sandy lake shores in NW Poland. Morphological traits and chlorophyll *a* fluorescence (Fv/Fm) were measured in 581 individuals of *Drosera*. *D. anglica* occupies the best-lit and warmest habitats, and also those that are the most heavily hydrated and the richest in organic matter; its rosettes are larger under conditions of higher pH, less organic matter, and less well-lit habitats. *D. intermedia* occupies substrates with the highest pH but the lowest conductivity, the poorest level of organic matter, and the least hydration. It is highly variable in terms of individual architecture. *D. rotundifolia* occupies habitats that are the most diverse, and that are often poorly lit, with the lowest pH but the highest conductivity. It is the least variable in terms of individual architecture. The value of the Fv/Fm ratio in *Drosera* is low (0.616 ± 0.137). The highest photosynthetic efficiency is achieved by *D. rotundifolia* (0.677 ± 0.111). It is significant for all substrates, indicating its high phenotypic plasticity. The other species have lower and similar Fv/Fm values (*D. intermedia*, 0.571 ± 0.118; *D. anglica*, 0.543 ± 0.154). Due to its very low photosynthetic efficiency, *D. anglica* avoids competition by occupying highly hydrated habitats. *D. intermedia* has adapted to the occupation of highly variable habitats in terms of hydration, while *D. rotundifolia* is primarily adapted to variable light conditions.

## 1. Introduction

There are only three species of *Drosera* spp. among the European flora: *D. rotundifolia* L., *D. anglica* Huds., and *D. intermedia* Hayne. They are included in the temperate perennial *Drosera* group [1]. Two natural hybrids are also encountered: (1) the fairly common *D*. × *obovata* Mert. et W.D.J.Koch—a hybrid of *D. rotundifolia* and *D. anglica*—and (2) found in Western Europe, *D*. × *beleziana* Camus—a hybrid of *D. rotundifolia* and *D. intermedia* [2] (the name *D*. × *eloisiana* has also been proposed [3]).

Due to anthropogenic pressure and the action of adverse environmental factors, naturally occurring plants are constantly exposed to functioning under suboptimal conditions. The ability of plants to function effectively and to survive under these adverse and changing habitat conditions determines the survival of the plants in the wild. The presence and intensity of environmental stress also affects the physiological states of plants in crops, where effective management requires in vivo monitoring of the physiological states of plants, their responses to stress, and their ability to maintain a high degree of productivity under adverse conditions. For this reason, a number of techniques have been developed to assess the physiological states of plants, among which are methods for estimating photosynthetic efficiency, which is the basis for the functioning of autotrophic organisms, which are at the forefront [4]. These methods are used both in typical agricultural and forestry crops, as well as in biomonitoring and in studies of the photosynthetic activities of plants in natural areas, such as protected areas. One of the methods mentioned is the analysis of the induction and the extinction curve of chlorophyll *a* fluorescence in vivo (the OJIP test) [5,6]. This is a rapid diagnostic method for estimating the efficiency of photosynthesis, and for detecting disorders of this process caused by environmental stress, often leading to damage to the photosynthetic apparatus [5,7]. The OJIP test has been successfully used to compare the photosynthetic performances of different plant species, individuals of the same species growing under different environmental conditions, and plants exposed to stress factors such as light, thermal, water, or chemical stress [6,8,9,10]. Importantly, plants and their parts are not destroyed during the measurements described above, which makes it possible to work with protected species growing in their natural habitats, such as swamp plants of the genus *Drosera*. One of the parameters determined using this method is the Fv/Fm ratio, which is recognized as a reliable measure of the photochemical activity of the photosynthetic apparatus. This is the maximum photochemical yield of PSII, which under optimal conditions of plant growth should be approximately 0.85 relative units [11]. A decrease in the value of this parameter indicates the occurrence of stress, which can lead to the phenomenon of photoinhibition. Very low values of this parameter (about 0.2–0.3) indicate irreversible changes in the structure of PSII.

From the literature, we know what types of habitats are occupied by each sundew species, while there are still little data on the values of individual abiotic environmental characteristics, especially those factors that affect the photosynthetic efficiency of sundews. The purpose of this study is to determine the individual architectures and the values of the Fv/Fm ratio in three *Drosera* species and to identify the features of the abiotic environment that affect them. It is intended that these studies are conducted in natural and untransformed habitats of *D. rotundifolia*, *D. anglica*, and *D. intermedia* within protected sites to avoid anthropogenic influence. In addition, it is intended that the differences between the Fv/Fm of the species are determined, as well as the types of substrates that they inhabit.

## 2. Results

### 2.1. Environmental Conditions of Sundews

The sundews studied are found in very well-lit habitats, on highly acidic substrates that are highly hydrated and rich in organic matter, as characterized by a low degree of electrolytic conductivity. Despite their occurrence within the same bog, the sundews studied show great habitat variation. *D. anglica* occurs primarily in the shore zone of dystrophic lakes (66.4% of the studied individuals; Table 1), sometimes on living *Sphagnum* mosses (19.2%) or in exposed peat (14.4%). It occupies the best-lit and warmest habitats among the sundews, and also the habitats that are most heavily hydrated and the richest in organic matter (Table 2, Figure 1). *D. intermedia* prefers a substrate that is distinguished by the highest pH (4.78 ± 0.66) but the lowest electrolytic conductivity; moreover, its habitats are the poorest in organic matter and the least hydrated. It is noteworthy for *D. intermedia* that apart from bogs, where it grows mainly in water (36.7% of individuals) and less frequently on peat and *Sphagnum* (6.7% each), it is very common and abundant on the sandy shores of oligotrophic lakes (50% of studied individuals). On such mineral substrates that are usually poorly hydrated, there is also *D. rotundifolia* (19.6% of individuals), which in peatlands occupies mainly *Sphagnum*-dominated habitats (63.0), as well as peat (10.9%), and much less frequently, water (6.5% of individuals, three sites—Table 1). The round-leaved sundew occupies habitats that are the most diverse in terms of characteristics (Figure 1), with substrates that are often poorly lit and that are distinguished by the lowest pH and the highest electrolytic conductivity.

*Drosera* species occupy diverse habitats within the studied bogs (Appendix A). It is noteworthy that *Drosera anglica* is the only one that does not occur on the mineral shores of oligotrophic lakes, and so it always occupies organic and highly hydrated habitats. Factor analysis showed that *D. anglica* habitats are mainly differentiated by the electrolytic conductivity of the substrate, PAR light intensity, and pH—environmental characteristics that are most strongly correlated with Factor 1 (the correlation coefficients are −0.85; −0.74, and 0.74, respectively). Substrate temperature is most positively correlated with Factor 2 (0.72). *D. intermedia* habitats are differentiated primarily by substrate hydration and organic matter content (strongly negatively correlated with the first factor, respectively: −0.95 and 0.96), as well as pH (positively correlated, 0.83), and to a slightly lesser extent, electrolytic conductivity (negatively correlated with the second factor, −0.76). As in the case of *D. intermedia*, the habitat of *D. rotundifolia* is also differentiated primarily by substrate hydration and organic matter content (strongly negatively correlated with the first factor; respectively: −0.91 and 0.93), but also by PAR light intensity and temperature (positively correlated with the second factor, 0.86 and 0.75, respectively; Appendix A).

The variation in *Drosera* habitat conditions is clearly bidirectional (Appendix A). The first direction is determined by substrate hydration and organic matter content (correlated with t1: −0.58 and −0.61, respectively). This direction is indicated by two significantly different sundew habitats, i.e., highly hydrated and organic matter-rich bogs (left part of the diagram), and poorly hydrated and mineral-rich lake shores (right part of the diagram). Attention is drawn to the occurrence of *D. anglica* in the most heavily hydrated substrates of peatlands. The second direction of variation in habitat conditions is indicated primarily by conductivity (−0.67) and substrate pH (0.47), and to a lesser extent, by temperature (0.37) and PAR light intensity (0.36; correlated with t2). It is noteworthy that in terms of these traits, the habitats of *D. rotundifolia* are highly differentiated, especially within the peatlands (Appendix A). The other species show similar variations in terms of these traits.

### 2.2. Individual Architectures of Sundews

The individual architecture of each *Drosera* species clearly differs in almost all plant characteristics, except for the average number of flowers per inflorescence (Figure 2, Table 3). *D. anglica* is, of course, distinguished by its largest rosette size, but also by its shoot and root sizes. On the other hand, *D. intermedia* is characterized by the highest number of leaves in the rosette, both living and dead. It is worth noting that the maximum root length value was also found in *D. intermedia* growing in water. *D. rotundifolia*, despite its small rosette size, is able to produce an exceptionally long main shoot (up to 5.73 cm), especially for individuals growing on live *Sphagnum* mats. Some individuals of *D. rotundifolia* are able to produce as many as four inflorescences per growing season.

Intraspecific variation in *Drosera* architecture is high. It is related to the type of substrate occupied (Figure 3, Figure 4 and Figure 5). The most impressive individuals of *D. anglica* grow in water, while the smallest grow among *Sphagnum* mats (Figure 3). Peat is the most variable habitat during the growing season of those inhabited by individuals of this species, especially in hot weather, when the black surface of the exposed peat heats up strongly. The plants here are most strongly colored red; their leaves age and die quite quickly.

*D. intermedia* occupies both aquatic habitats, peat and sand. On sandy substrate, individuals are stunted, rosettes are very small but form distinct clumps characterized by a large number of living leaves and inflorescences (Figure 4). Additionally, growing in water, *D. intermedia* often forms clumps, but of much larger size than on sand.

In the case of *D. rotundifolia*, the individuals are very similar to each other, regardless of the type of substrate on which they occur (Figure 5). Growing on sand, the round-leaved sundew produces a lot of inflorescences and leaves, while in compact *Sphagnum* mats it does not form the rosette typical of the species, and the main stem is strongly elongated (Figure 5).

### 2.3. Influence of Environmental Factors on Individual Architecture

Correlation analysis showed that only a few architectural traits of sundew individuals are strongly correlated with environmental traits. Many of these correlations are statistically significant, but they tend to be very weak. The length of sundew roots is positively correlated with substrate hydration (r = 0.48) and organic matter content (r = 0.49), while the number of live leaves is positively correlated with pH (0.48) and negatively correlated with electrolytic conductivity (−0.41), as well as substrate hydration (r = −0.43) and organic matter content (r = −0.47).

In general, no more correlations were found between the environmental characteristics and individual architectures when examining relationships within species. In the case of *D. anglica*, rosette size is positively correlated with substrate pH (0.43). This is confirmed by multivariate analysis (Figure 6), which further indicates that *D. anglica* rosettes are larger in less organic and less well-lit habitats.

The length of the roots of *D. intermedia* is positively correlated with organic matter content (r = 0.77) and substrate hydration (r = 0.75), and negatively correlated with pH (−0.49). The main axis height of *D. intermedia* is also correlated with organic matter content (r = 0.52) and substrate hydration (r = 0.49). RDA analysis additionally indicates a negative effect of several environmental traits, including PAR light intensity, organic matter content, and substrate hydration on the number of living leaves (cf. Figure 6), which is primarily associated with individuals found on mineral lake shores.

For *D. rotundifolia*, strong correlations relate to the positive effects of organic matter content (r = 0.49) and substrate hydration (r = 0.45) on root length. According to RDA analysis, this is also the case for the length of the main shoot. The number of live leaves is negatively affected by organic matter content and substrate hydration (−0.57 and −0.50, respectively), indicating that a large number of leaves in the rosette of this sundew will be characteristic of individuals found on mineral and poorly hydrated substrates (sandy lake shores), as is the case with *D. intermedia*.

### 2.4. Photosynthetic Efficiency

The Fv/Fm ratio for *Drosera* averages 0.616 ± 0.137 (Table 4) and varies slightly between species (Figure 7). The highest values are obtained by *D. rotundifolia* (0.677 ± 0.111); the other species have lower Fv/Fm (*p* < 0.001) values, which are close to each other and not statistically different (*p* > 0.05; Table 4). It is worth noting that the range of variability for each species is very wide. The correlations of Fv/Fm of species with the environmental traits found in the study are weak, and only a few of them are statistically significant (Table 5). This is due to the high variability of the habitats occupied by individuals of each species, as each species is usually found within a single bog, both in water, on peat, and in dense mats of *Sphagnum*.

*D. anglica* achieves the highest values of the Fv/Fm index when growing in dense *Sphagnum* communities (Figure 8), and they are higher, especially in relation to those found in individuals growing on peat (*p* < 0.001), and to a lesser extent, to those growing in water (*p* = 0.017). Plants achieve high Fv/Fm values, especially when the substrate conductivity is high, and when pH and substrate hydration are low, as evidenced by strong correlations with these traits (Table 6). Significantly lower Fv/Fm values are obtained by *D. anglica* growing in water, where temperature and PAR light intensity have the greatest influences on this value. This species achieves the lowest Fv/Fm values when growing on peat, especially peat with the lowest conductivity and a low temperature, the highest pH, and a high organic matter content.

*D. intermedia* achieves the highest values of the Fv/Fm index in extremely different habitats, i.e., in water and on piscina, and these values are higher than those obtained by individuals growing on *Sphagnum* mats (*p* = 0.014 and *p* = 0.017, respectively). It is worth noting, however, that in these habitats, different environmental characteristics determine the value of the Fv/Fm index. In poor water, it is favorably affected by high conductivity and PAR intensities, while on sand, it is primarily affected by the high organic matter content of the substrate. In both cases, a low substrate pH is beneficial. In other habitats, *D. intermedia* is found rarely, and the effect of environmental conditions on Fv/Fm cannot be interpreted. It achieves the lowest values when growing on live *Sphagnum*.

*D. rotundifolia*, in contrast to other species, obtains the highest Fv/Fm values on peat (Figure 8), especially habitats distinguished by high conductivity, poor light, poor hydration, and poor organic matter levels (Table 6). Growing on sand, it obtains only slightly lower Fv/Fm values, and it is favorably influenced by hydration, but also by a low pH and a low PAR light intensity. Importantly, *D. rotundifolia* also obtains a fairly high Fv/Fm ratio when growing on other substrates: on *Sphagnum* mats, especially when the substrate is rich in organic matter, and in water when it has a low electrolytic conductivity (Table 6).

Our studies show that in the case of *D. anglica*, the PAR irradiance has no effect on the maximum quantum yield of photosynthesis; this sundew is always found in highly lit habitats and obtains both high and very low Fv/Fm ratio values under such conditions. *D. intermedia* and *D. rotundifolia* obtain high values of the Fv/Fm ratio at a low PAR light intensity (Figure 9), within 60–80% of what reaches the bogs. Primarily, there are areas that are occupied by low-density rushes, or the juvenile stage of the swamp forest, in the case of *D. rotundifolia*. At PAR intensities of above 90%, the values of the Fv/Fm ratio decrease markedly, and the lowest Fv/Fm values of 0.2–0.3 are obtained by these sundews in areas with direct and full PAR light intensities (100%).

*D. anglica* is a species that is associated with the warmest habitats in peatlands, and we can conclude that a decrease in substrate temperature has a clear effect on the functioning of its photosynthetic apparatus, as it affects a significant reduction in the value of the Fv/Fm ratio (Figure 10). For the other sundews, it seems that temperature is not a stress factor, as it does not affect the Fv/Fm value. This is particularly evident for *D. rotundifolia*, as there were no statistically significant differences in the substrate temperature over the entire range of Fv/Fm variation (*p* = 0.68).

As with temperature, substrate hydration also does not significantly affect the value of the Fv/Fm ratio in *D. anglica* (Figure 11). In the case of the other sundews, the influence of substrate hydration is not clear, which is due to the occurrence of this species in heavily hydrated bogs, as well as in poorly hydrated sands. They obtain both high and low Fv/Fm values on the most poorly hydrated substrates, with a hydration of about 65% for *D. rotundifolia*, and a slightly lower one of about 50% for *D. intermedia*. On more strongly hydrated substrates, these sundews achieve intermediate Fv/Fm values (0.5–0.6).

## 3. Discussion

### 3.1. Variability of Individual Architectures and Photosynthetic Performance of Sundews

As typical carnivorous plants, sundews are found mainly in open and strongly sunny habitats that are warm or hot in summer, while also being highly hydrated, acidic, and nutrient poor, because only under such conditions they do have a significant competitive advantage over non-carnivorous plants [12]. They are a characteristic element of *Sphagnum*-dominated communities in these habitats [13]. Due to the adaptation of sundew leaves to carnivorousness—that is, to attract, capture, and digest prey, sundews have a reduced rate of photosynthesis and they are unable to function under low-light conditions—carnivory becomes too costly in such habitats [14]. The efficiency of photosynthesis, estimated from the parameters of chlorophyll *a* fluorescence in vivo, is specifically influenced by individual nutrients, one of the most important being nitrogen availability [15,16]. In general, plants respond to low nitrogen availability by redistributing it from the older leaves to the youngest [17], which obviously affects the photosynthetic functions of the leaf. Although carnivorous plants can function without prey capture [18], prey capture clearly increases plant growth, the abundance of flowering, and seed production [19,20,21].

The sundews, even within a single peatland, occupy very diverse habitats (cf. Figure 1). This diversity, as well as the rather specific and difficult habitat conditions in peatlands for plants, makes both the individual architectures of sundews and their photosynthetic performances highly variable. A good indicator of the state of the photosynthetic apparatus of plants under different environmental conditions is the Fv/Fm parameter, which indicates the maximum quantum yield of photosynthesis, as it depends mainly on the environmental factors that have a stressful effect on plants [22]. Fv/Fm values are relatively stable, while they begin to decline at levels indicating severe or extreme stress. According to many authors, a sustained decline in the Fv/FM parameter is a reliable indicator of photoinhibition [23] and indicates an early stage of leaf aging [24]. The result of this stress can be changes in individual plant morphological traits, such as size, the number of leaves produced, or the abundance of flowering and fruiting [25]. The detection of these morphological changes and their long-term monitoring is now becoming important due to global climate change.

It seems that the individual architectures of *Drosera*, as well as the maximum photosynthetic efficiency (Fv/Fm), are more heavily influenced by the type of habitat/substrate occupied than by single abiotic features (cf. Figure 3, Figure 4, Figure 5 and Figure 8), as it exhibits a complex interaction of several environmental features. Individual sundew species show a preference for specific habitats within the bog [26], and the architectures of their individuals indicate different adaptations to these conditions [27,28,29,30]. However, it is worth noting that even the same type of substrate often possesses different physical and chemical properties, e.g., water and peat differ between bogs [31], and similarly, a mat of *Sphagnum* is built by completely different species in clumps, in valleys, and still others in water; this results in the additional diversity of *Drosera* habitats [13,26,32].

Our study shows that individual environmental factors have a relatively weak but statistically significant effect on sundew traits (cf. Table 6). Higher substrate hydration and organic matter content contribute to sundews producing longer roots but producing fewer living leaves [28].

In the case of *D. anglica*, individuals are larger in habitats with a higher substrate pH, less organic matter, and less light (cf. Figure 6), and they generally comprise low rushes growing in dystrophic reservoirs (cf. Figure 3). The length of roots and the main axis height of *D. intermedia* are positively correlated with the organic matter content and substrate hydration (cf. Figure 4), while a large number of living leaves is negatively correlated with these characteristics. This is mainly due to the peculiarities of *D. intermedia* individuals growing on sand; they are characterized by a large number of leaves [26,28].

In the case of *D. rotundifolia*, we also found a clear positive effect of organic matter content and substrate hydration on root length, as well as on the main shoot length. As in *D. intermedia*, organic matter content and substrate hydration had a negative effect on the number of live leaves. A large number of leaves in a rosette is characteristic of individuals found on mineral and poorly hydrated substrates, where individuals also have a poorly developed (short) root system (cf. Figure 5).

The showiest individuals of *D. anglica* that grow in water do not obtain high Fv/Fm values, indicating that such a substrate can be a stress factor for plants [22,33,34]. Above all, it has the most well-lit and warmest habitat, and of course, the most well-hydrated. The highest quantum yields of photosynthesis are distinguished by individuals growing in compact sphagnum mats (cf. Figure 8) where conditions seem to be the most stable, and at the same time, the conductivity of the substrate there is relatively high, which ensures a significant availability of ions, while the pH and hydration of the substrate are lower compared to the other types of substrates (although the substrate is still strongly hydrated). *D. anglica* achieves the lowest values of the Fv/Fm ratio when growing on peat. This substrate is distinguished by the lowest electrolytic conductivity and a low temperature, while it has the highest pH and organic matter content. This is the most variable habitat during the growing season of those inhabited by individuals of this species, especially in hot weather, when the black surface of the exposed peat heats up strongly. The plants here are most strongly colored red, their leaves age and die quite quickly, and the plants produce turions early [23,24].

*D. intermedia* grows in acidic bogs in parts of valleys that are flooded in winter and that often dry out in summer. It is also widespread in permanently flooded areas, as is the case in the peatlands that we studied. *D. intermedia* is described as the sundew of the most heavily hydrated habitats in peatlands, and it is often found in water, and even underwater [27]. Importantly, *Drosera intermedia* is the only one that can tolerate long periods of submersion. Plants that grow in water produce longer stems and long petioles. Our research confirms this; *Drosera intermedia* in water hits very often and reaches its largest size here (both the rosette size and root length). Individuals of these habitats also produce a large number of leaves and produce many showy inflorescences, which are more robust than those of *D. rotundifolia* [27], and on many individuals, the inflorescences from the previous growing season persist.

The adult plants of this species can also tolerate areas of peatland that dry out slightly in the summer months [28], which is not confirmed by our study, since this sundew is found only in the central parts of peatlands that are the most highly associated with heavy hydration, and often with open water, where it also obtains the highest values of the Fv/Fm ratio. It obtains similar values when growing in extremely different habitats, namely sand. *D. intermedia* achieves the lowest Fv/Fm values when growing on live sphagnum mats. In poor water, the index is favorably affected by high conductivity and high PAR intensities, while on sand, it is primarily affected by the high organic matter content of the substrate. On both substrates, the low pH of the substrate is beneficial.

*D. rotundifolia*, in contrast to the other species, obtains the highest photosynthetic quantum yield on peat (cf. Figure 8), which is distinguished by a high conductivity for these habitats, but by low light and poor hydration. These are very often habitats that are subject to rapid succession, and in which there is considerable shading of sundews (usually a young swamp forest with a high abundance of shrubs), but in extreme shade, its leaves become more markedly etiolated, the growth weakens, and the plants may die. Among European species, *D. rotundifolia* tends to grow in the highest areas of peatland microrelief [30], and primarily on tall *Sphagnum* clumps [13,32], which may indicate a clear preference for open and sunny habitats. However, in dense *Sphagnum* mats, it does not form a ground-level rosette typical of the species, and the main stem is strongly elongated (cf. Figure 5), as the plant must grow relatively quickly because it is in competition with fast-growing *Sphagnum* mosses [27,35].

Growing on sand, the round-leaved sundew produces a lot of inflorescences and leaves, but it achieves slightly lower Fv/Fm ratio values here. The presence of *D. rotundifolia* and the attainment of high and similar Fv/Fm values, both in full light and shade, indicates the high photosynthetic plasticity of this species. This observation is consistent with the results presented by Bruzzese et al. [36], indicating the high photosynthetic rates of *D. rotundifolia* compared to *D. capensis* and *Sarracenia leucophylla*. Studies also indicate that *D. rotundifolia* is highly adapted to the microclimatic conditions of peatlands [37], including high humidity, frequent fog and dew, more frequent frost, and high summer temperatures [38]. The species avoids wet places with standing water [39], but in our study, there were individuals growing in water in a slight dip at the edges of dystrophic ponds. Their architecture (cf. Figure 5) does not differ from that found in other habitats, but the Fv/Fm ratio values are the lowest, suggesting highly stressful conditions for *D. rotundifolia*. Growing in habitats that are rich in nitrogen and phosphorus, sundews reduce their rosette size while producing fewer leaves and flowers [40].

### 3.2. Effect of Light Stress on the Maximum Quantum Yield of Photosynthesis of Drosera

Many carnivorous plants exhibit high ecophysiological plasticity [41] and can grow at relatively low as well as high solar irradiances. Our studies show that in the case of *D. anglica*, the PAR irradiance has no effect on the maximum quantum yield of photosynthesis; this sundew is always found in highly lit habitats and obtains both high and very low Fv/Fm ratio values under such conditions. *D. intermedia* and *D. rotundifolia* obtain high values of the Fv/Fm ratio at a low PAR light intensity (cf. Figure 9).These results are consistent with reports that in carnivorous plants, the photosynthetic light saturation point reaches relatively low values [36]. This indicates that excessive light in *D. intermedia* and *D. rotundifolia* can interfere with photosynthesis, or in extreme cases, it can cause damage to the photosynthetic apparatus, which is often observed in plants [42]. Plants have developed a wide variety of mechanisms and adaptations to tolerate excessive PAR light intensity, or to nullify the effects of light stress [43,44,45,46]. During prolonged sun exposure, plants acclimate to the conditions, reducing light absorption and increasing photoprotection against excessive light and its harmful effects. The results of Tkalec et al. [47] strongly suggest that sundews are able to successfully acclimate to both low and high light intensities by changing the content and composition of photosynthetic pigments and phenolic compounds. Plants growing under low-light conditions adapt to high-intensity sunlight by increasing the content of phenolic compounds, especially anthocyanins, and decreasing the content of photosynthetic pigment. It is possible that *D. anglica* has developed such adaptations, and in this case, the values of the Fv/Fm ratio are determined by environmental characteristics other than the PAR light intensity.

### 3.3. Effect of Temperature on the Maximum Quantum Yield of Photosynthesis of Drosera

Plants in the temperate climate zone are subject to both low- and high-temperature stresses resulting from significant fluctuations in ambient temperature [48]. In particular, in relatively cold habitats such as peatlands, the summer temperatures are extremely high, especially on exposed peat or in shallow water bodies that are rich in humic substances [49]. There are numerous examples of the effects of high and low temperatures on the photosynthetic activity of leaves, as expressed by different parameters of chlorophyll *a* fluorescence in vivo [50,51].

At low temperatures, many changes occur in plants, leading to a reduction or an inhibition in photosynthesis [52], which is reflected in the values of chlorophyll *a* fluorescence parameters in vivo [53]; the probability of photoinhibition then increases significantly [54].

Our studies show that *D. anglica* is a species that is associated with the warmest habitats in peatlands (cf. Figure 1), and we can conclude that a decrease in substrate temperature has a clear effect on the functioning of its photosynthetic apparatus, as it affects a significant reduction in the value of the Fv/Fm ratio (cf. Figure 10). For the other sundews, it seems that temperature is not a stress factor, as it does not affect the Fv/Fm value. Thus, the studied species are well adapted to both low and high temperatures in peatlands, and it seems that temperature is not a stressor in most *Drosera* habitats.

As hemicryptophytes with hardy hibernacula (winter buds), sundews survive winter and spring frosts well [55]; in the case of *D. rotundifolia*, generally in *Sphagnum* mats, and in the case of the other species, mostly underwater.

In response to high-temperature stress, there are also changes in the chlorophyll *a* fluorescence parameters in vivo [56]. High-temperature stress causes many changes in plants, including changes in the permeability of the thylakoid membranes, as well as changes in the conformation of membrane proteins [57], and it causes the formation of membrane monolayers [58]. 

### 3.4. Effect of Water Stress on the Maximum Quantum Yield of Photosynthesis of Drosera

Water stress is primarily considered in plants as a shortage of water, but in the case of a specific group such as sundews, water stress can also be considered as excess water that is associated with the flooding of plants, or permanent submergence. Each sundew species tolerates these periods of stress differently. We know from the literature that *D. intermedia* best tolerates periods of drought in peatlands, and our research confirms that it also does very well growing in water, as well as underwater as a permanently submerged plant. In the case of other species, the submergence of sundews can be only rather short-lived; over the long-term, it quickly results in the death of the plants, although they grow quite well in water. They are usually found on submerged *Sphagnum* mats at the edges of mid-peat ponds; one also finds them on peat exposed after the water level drops, or on mats of living *Sphagnum*. These habitats are considered as an advanced stage of aquatic habitat succession, where substrate hydration is still very high and constant, and very rich in organic matter. This species tolerates periodic submergence, but not dry or shaded conditions [26]. *D. rotundifolia* is found quite often in poorly hydrated habitats; it tolerates periods of drought well if they are not prolonged, and its sites are shaded.

Substrate hydration does not significantly affect the value of the Fv/Fm ratio in *D. anglica* (cf. Figure 11). This sundew is always found in heavily hydrated sites, so it does not function well under water stress. In the case of the other sundews, the influence of substrate hydration is not clear, which is due to the occurrence of this species in heavily hydrated bogs, as well as in poorly hydrated sands (cf. Figure 1). High Fv/Fm values on poorly hydrated mineral habitats (sandy lake shores) indicate that water stress conditions are not present for *D. intermedia* and *D. rotundifolia*.

Water deficiency reduces the rate of photosynthesis, and thus it can inhibit plant growth and productivity [23,59]. Some researchers claim that PSII is quite resistant to water stress, and that the effect of water stress on photochemical responses manifests itself only under severe drought stress [60], which is usually not the case for sundews found in untransformed peatlands.

Water deficit can impede gas exchange and thus slow down the rate of CO_2_ assimilation, which is associated with the closure of stomata and changes in the rates of various metabolic reactions [23]. However, some authors believe that this has little effect on plants under mild to moderate drought [61]. Under natural conditions, however, water stress is often associated with a high irradiance level and a high temperature. The co-occurrence of various abiotic stressors contributes to significant changes in chlorophyll *a* fluorescence kinetics in vivo, reduced photosynthetic efficiency, and chronic photoinhibition [62]. Additionally, such a co-occurrence of various abiotic stresses can be attributed to the very low Fv/Fm values of *Drosena anglica* and *D. intermedia* in some habitats, such as exposed peat (cf. Figure 8).

In the case of severe drought, the decrease in maximum PSII quantum yield (Fv/FM) is very pronounced [60], but it depends on the time after which rehydration occurs (if any), as well as on the plant species, since they show different tolerances of the photosynthetic apparatus to a lack of water [63]. In the case of sundews, this problem can practically only affect highly transformed habitats. It is worth noting, however, that the exposure of a plant to water stress can alter its responses to higher temperatures.

Stress caused by excessive soil moisture has an enormous impact on wild plants and agricultural crops [64]. A high level of soil moisture retards the growth of many plants, especially in humid regions [65]. Flooding changes the physical, chemical, and biological parameters of the substrate, which significantly affects the plant growth conditions. Inundation reduces the lengths of the shoots and roots of terrestrial plants, reduces the total biomass, causes adverse changes in biomass allocation, and stimulates plant senescence [66]. However, for swamp plants, most of these changes are not found. Our studies show that in *Drosera*, there is a marked elongation of stems and roots, and the overall biomass (size) of the plants increases. This is corroborated by studies of other swamp plants (*Phragmites australis*, *Carex cinerascens*, and *Hemarthria altissima*). None of these species showed significant changes in PSII activity, indicating that the photosynthetic apparatus was not significantly damaged under substrate flood stress [67]. *D. intermedia* is the species most tolerant to overwatering; it withstands flooding very well and is able to survive for long periods under complete submergence.

## 4. Methods

### 4.1. Collection of Samples and Methods of the Analysis of Environmental Conditions

Material for the study was collected during August and early September 2022 from 10 sites (Table 1, Figure 12) where sundews occurred. These were mainly well-preserved, raised, and transitional bogs (seven sites); and the sandy shores of oligotrophic lakes (three lobelia lakes), which are under legal protection, and mainly in reserves. Surveys were made of the individuals of each sundew species and the environmental conditions in which they occurred.

Individuals of three sundew species were collected for the study: *Drosera rotundifolia* L., *Drosera anglica* Huds., and *Drosera intermedia* Hayne. Within each study site, five survey sites were designated for each *Drosera* species found. At each such site, environmental conditions were determined and the architectures of six individuals of a given sundew species were measured. For this purpose, plants were obtained: they were plucked by hand from *Sphagnum* mosses and peat, taken out of water, or dug with a shovel from mineral sand without destroying the roots. The fluorescence of each individual was measured using a chlorophyll meter. All individuals from the site were carefully spread out on a special scaled backing and photographed for later measurements of architectural features using graphic programs. After the photos were taken, the plants were planted back in the same location from which they were taken. Three typical individuals of each sundew species were preserved as herbarium material in the Herbarium Universitatis Gedanensis UGDA. Using CorelDRAW X6 software from the photos, the architectures of the plants was measured, including: the height of their main shoot, rosette width and root length, the number of living and dead leaves, the number of inflorescences and flowers, and/or fruits on the inflorescence. In this way, results were obtained for 30 plants of each *Drosera* species found within each peat bog. In total, the characteristics of 125 individuals of *D. anglica*, 180 of *D. intermedia*, and 276 of *D. rotundifolia* were measured.

Using field meters, environmental characteristics such as (1) PAR light intensity, (2) substrate temperature, (3) substrate pH, and (4) electrolytic conductivity were measured at all sites at each study object. In addition, the type of substrate was determined within four categories: peat, water, sand, or live *Sphagnum* mosses (Table 1). A small sample of the substrate (about 250 mL) was also collected from each site for laboratory analysis, to determine its hydration and organic matter content.

The characterization of the environmental conditions at each site was carried out based on the following measurements and analyses:-Photosynthetically active radiation (PAR) intensity was measured with an LI-250 light meter, and an average of five measurements at the sundews was taken, which was then converted for the plant site as a percentage of light reaching the site/peatland in a fully illuminated location;-pH and temperature—with a WTW 320/SET1 pH meter and a SENTIX 97T measuring electrode;-Electrolytic conductivity—with a WTW Cond 3210 SET 2 conductivity meter;-Substrate hydration—calculated as a percentage from the difference in the weight of fresh substrate, and the substrate dried to constant weight at 105 °C in a Binder FD115 laboratory dryer;-The organic matter content of the substrate was calculated as a percentage from the difference in weight between the dry substrate and the substrate roasted in a SEL 96C muffle furnace at 550 °C for 5 h.

The chlorophyll *a* fluorescence of all individuals was measured using a Handy Pea fluorometer (Hansatech Ltd., Pentney, UK), as described by Aksmann et al. [68]. The photosynthetic efficiency was measured on the second or third youngest leaf of the plant. In the following work, only one of the measured parameters, Fv/Fm, an index of the maximum photochemical efficiency of the photosynthetic apparatus, was included in the analysis, where:-Fv/Fm—maximum photochemical quantum yield of photosystem II (PSII);-Fv—fluorescence variable (Fm − Fo);-Fm—maximum fluorescence (fluorescence level when all photochemical traps are closed);-Fo—minimum fluorescence (fluorescence level when all photochemical traps are open).

### 4.2. Statistical Methods Used in Developing the Results

The obtained results for both the plant measurements and the environmental conditions were summarized in a Microsoft Excel spreadsheet. Each individual was assigned appropriately characterized environmental conditions at the site, i.e., sunlight and substrate characteristics: pH, conductivity, temperature, hydration and organic matter content, and substrate type. Measurements of environmental conditions were made at 105 sites. The architectures of 585 individuals were determined, and photosynthetic efficiency was measured for each.

Using Statistica 13.1 software, the arithmetic mean and median, standard deviation, the minimum and maximum values of all plant traits, and the environmental conditions were calculated. Graphs were also created in this software to compare the traits of each species, and differences were determined according to ANOVA and Tukey’s post hoc test for unequal abundances. Significant differences at a probability level of *p* < 0.05 according to the Compact Letter Display (CLD) methodology were presented in graphs using different letters (a, b, c). The variation in environmental conditions among the sundew species was determined using Principal Component Analysis (PCA) and Partial Least Squares (PLS) analysis in Statistica 13.1. Relationships between individual architecture and environmental traits and photosynthetic performances were determined using Redundancy Analysis (RDA) in Canoco 5.1 [69], and the linear correlation was determined in Statistica 13.1.

## 5. Conclusions

The largest size is, of course, reached by *D. anglica*, but the individual architectures vary considerably, depending on the nature of the habitat. This is the species that is most strongly associated with aquatic habitats that are open and sunny, and the most heavily hydrated. *D. intermedia* is highly variable in terms of individual architecture, and it also occupies a wide variety of habitats in terms of hydration and organic matter content, from heavily hydrated peats, where it grows in and under water, to mineral substrates that are poorly hydrated. *D. rotundifolia* is the least variable in terms of rosette morphology, but it is best adapted to different environmental conditions, especially light.

The maximum quantum yield of photosynthesis of European *Drosera* is low (0.616 ± 0.137); the highest is obtained by *D. rotundifolia*, with the other species being slightly lower and similar. Many environmental factors affect the value of this indicator, and the most important should be considered: light intensity, temperature, and the availability of water, as well as nutrients.

Permit numbers obtained at the Regional Directorate of Environmental Protection in Gdansk.

Permission for sundew collection and environmental studies: RDOŚ-Gd-WZG.6400.98.2022.SK.2.

Consents for research carried out in reserves:RDOŚ-Gd-WOC.6205.46.2021.MaK.3—Żurawie Błota Nature Reserve;RDOŚ-Gd-WOC.6205.36.2022.JK.2—Żurawie Błota Nature Reserve, Krasne Lake Nature Reserve, Lisia Kępa Nature Reserve;RDOŚ-Gd-WOC.6205.28.2022.ATP.2—Małe Łowne Nature Reserve, Moczadło Lake Nature Reserve.

## Figures and Tables

**Figure 1 ijms-24-09823-f001:**
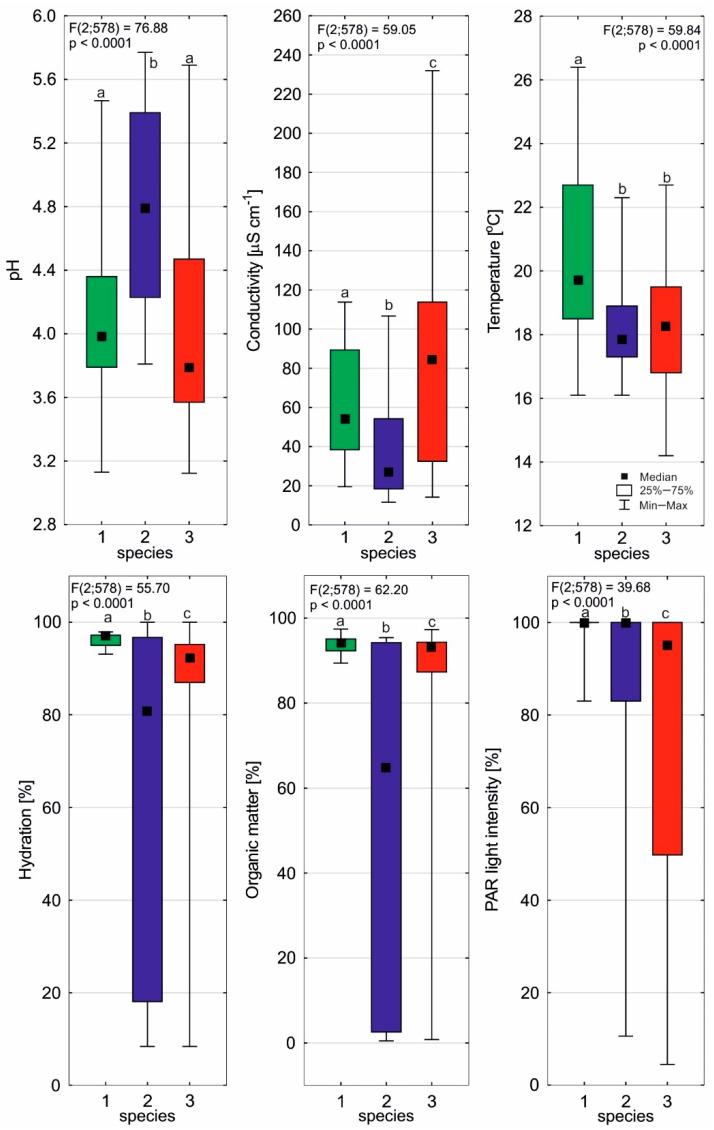
Environmental conditions of *Drosera anglica* (1), *D. intermedia* (2), and *D. rotundifolia* (3). Different letters (a, b, c) represent significant differences at *p* < 0.05 probability level according to RIR Tukey’s post hoc test using the Compact Letter Display (CLD) methodology.

**Figure 2 ijms-24-09823-f002:**
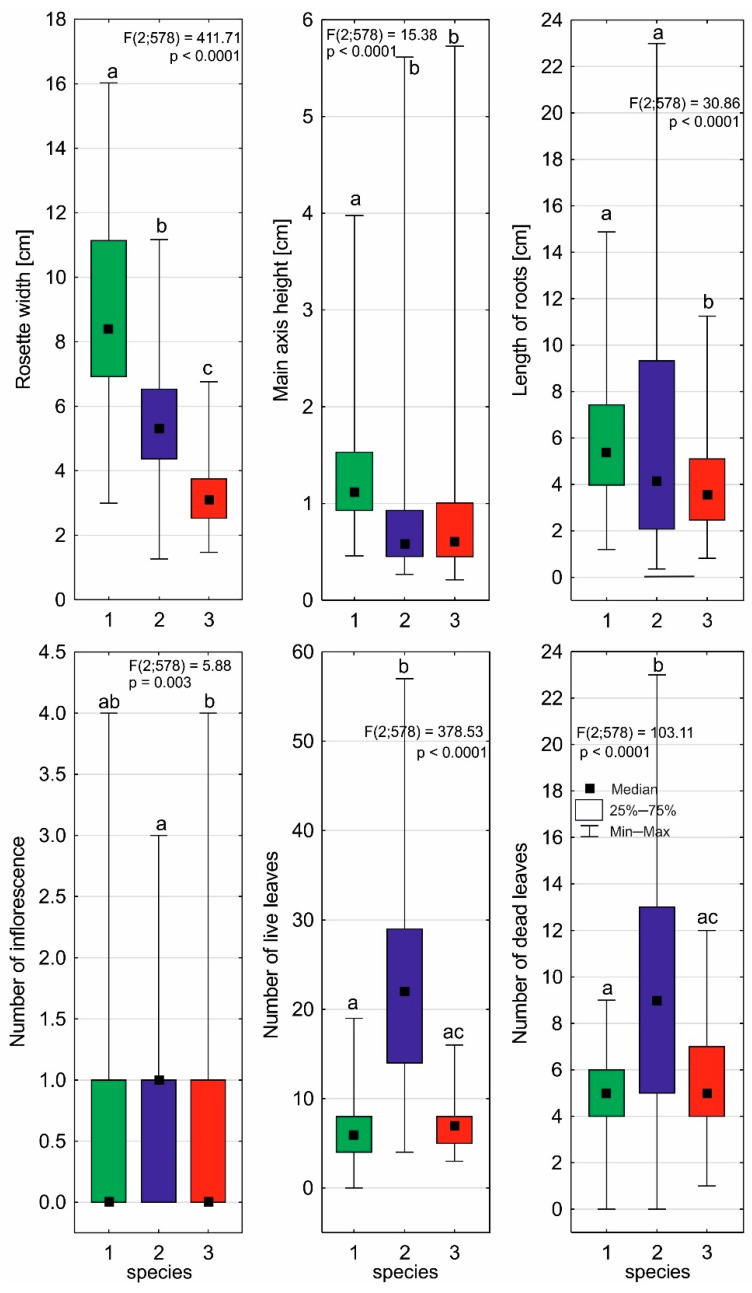
Differences in individual architectures of *D. anglica* (1), *D. intermedia* (2), and *D. rotundifolia* (3). Different letters (a, b, c) represent significant differences at *p* < 0.05 probability level according to RIR Tukey’s post hoc test using the Compact Letter Display (CLD) methodology.

**Figure 3 ijms-24-09823-f003:**
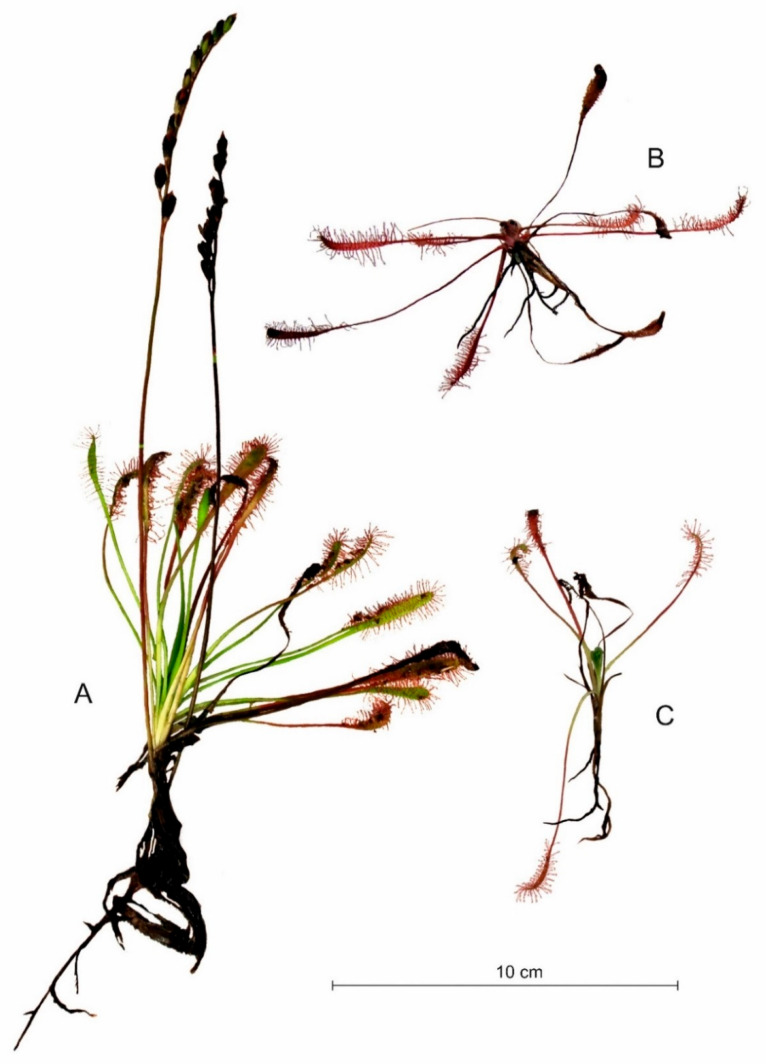
Characteristic individuals of *Drosera anglica* found in water ((**A**); shore of a mid-peat lake near Studzienice), on peat (**B**), and on a *Sphagnum* mat ((**C**); peatbog near Żuromino).

**Figure 4 ijms-24-09823-f004:**
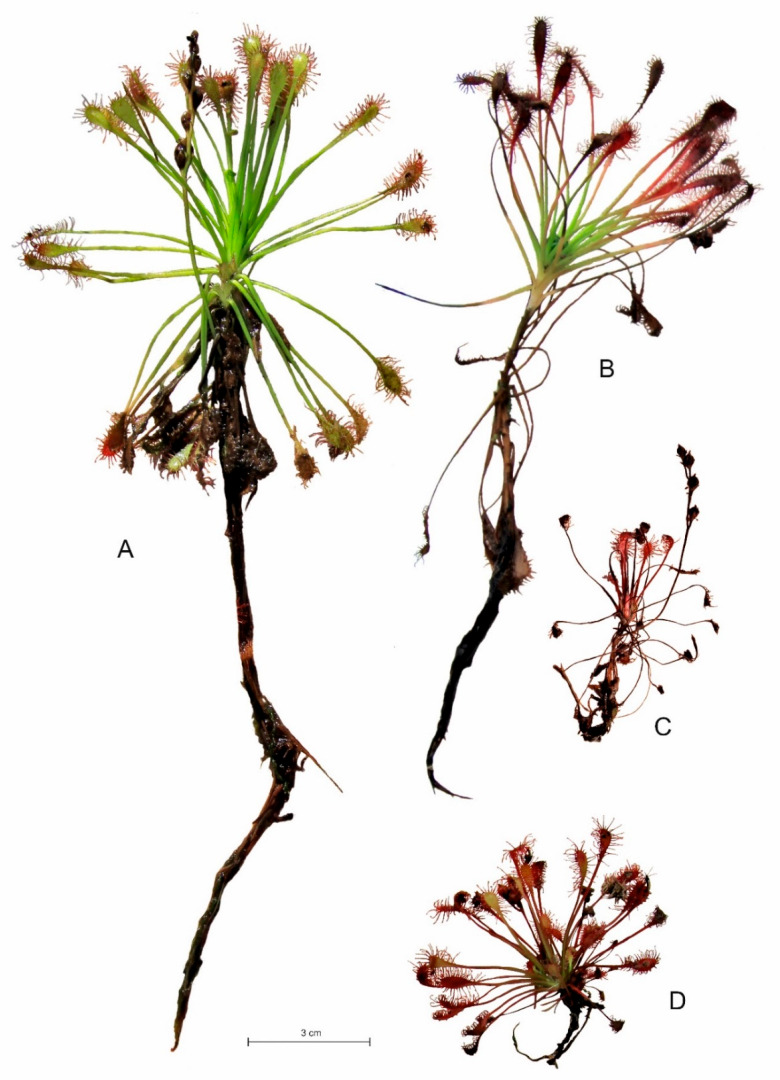
Characteristic individuals of *Drosera intermedia* found in water ((**A**); Studzienice), on peat ((**B**); Męcikał), on *Sphagnum* mats ((**C**); Leniwe Lake), and on sand ((**D**), shore of Krasne Lake).

**Figure 5 ijms-24-09823-f005:**
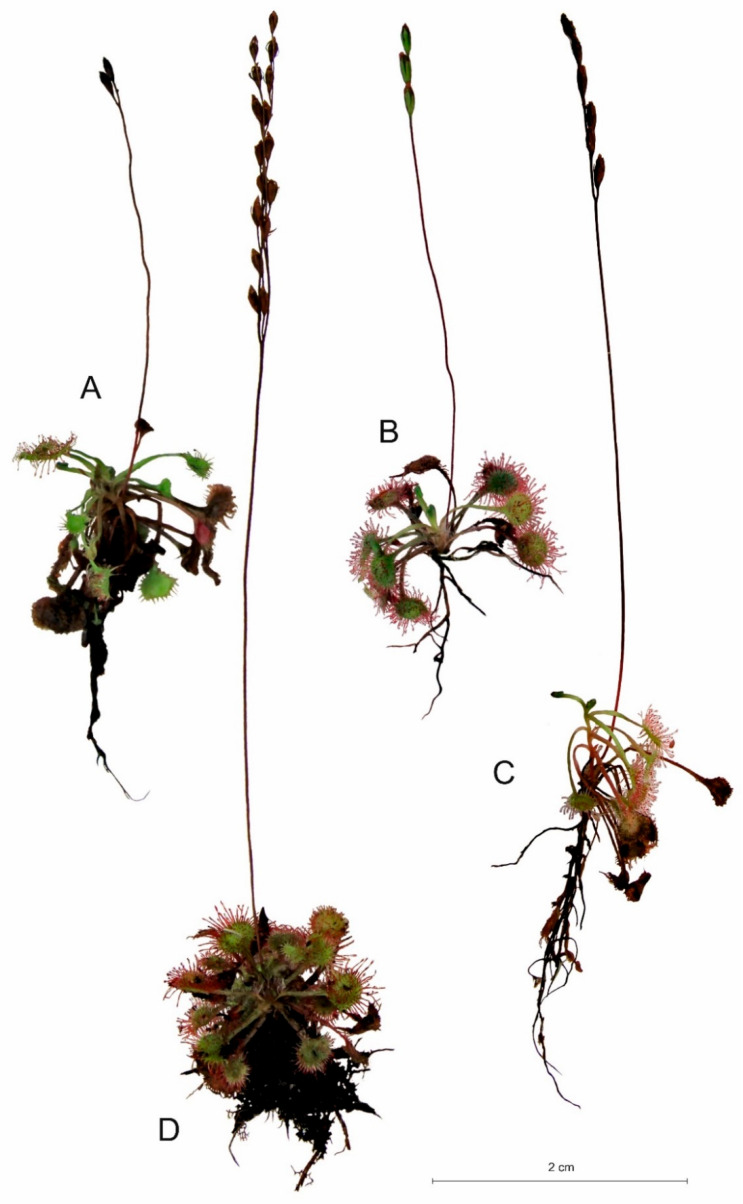
Characteristic individuals of *Drosera rotundifolia* found in the water ((**A**); Męcikał), on peat ((**B**); Zawiat), on sand ((**C**); the shore of Lake Moczadło), and on *Sphagnum* mats ((**D**); Lake Małe Łowne).

**Figure 6 ijms-24-09823-f006:**
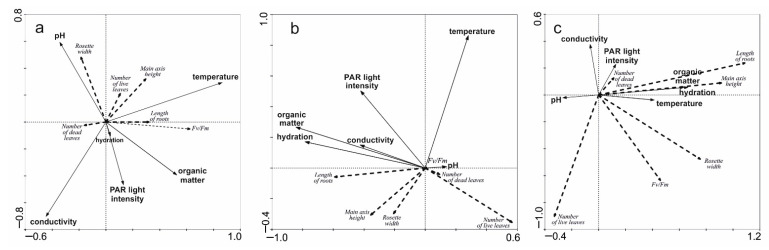
Effects of environmental characteristics on the architectures of *D. anglica* (**a**), *D. intermedia* (**b**), and *D. rotundifolia* (**c**) specimens, according to RDA analysis.

Summary of RDA analysis
*D. anglica**D. intermedia**D. rotundifolia*Axes121212Eigenvalues0.1080.0640.2370.0640.1410.052Species–environment correlations0.7140.5580.8250.5470.7610.477Cumulative percentage varianceof species data10.817.223.730.114.119.3of species-environment relation51.681.967.385.663.486.8Sum of all canonical eigenvalues0.2100.3510.223

**Figure 7 ijms-24-09823-f007:**
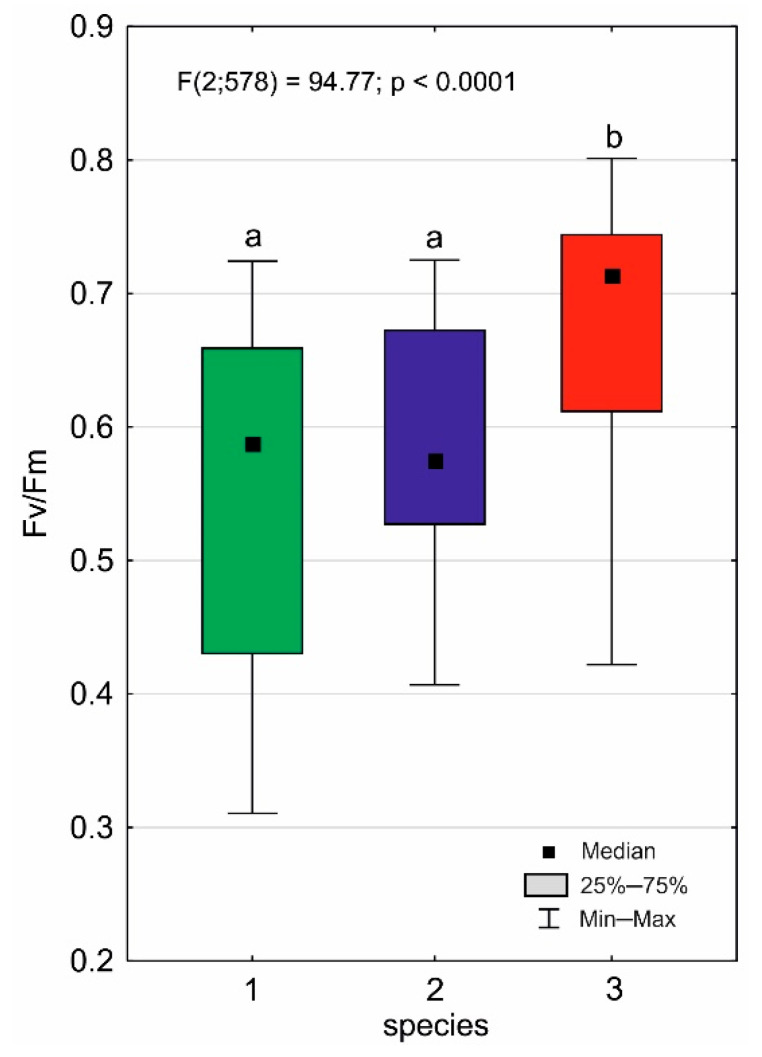
Fv/Fm ratios for individual *Drosera* species (1—*D. anglica*, 2—*D. intermedia*, 3—*D. rotundifolia*). Different letters (a, b) represent significant differences at *p* < 0.05 probability level according to RIR Tukey’s post hoc test using the Compact Letter Display (CLD) methodology.

**Figure 8 ijms-24-09823-f008:**
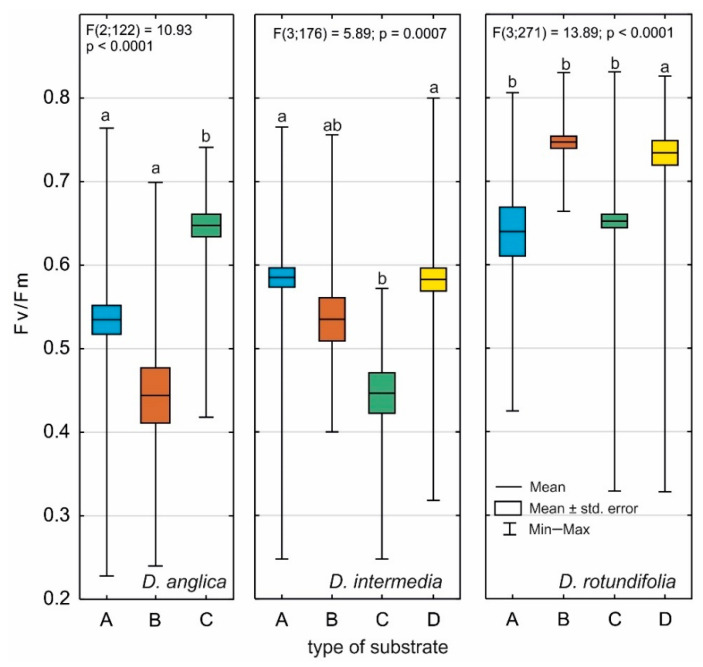
Fv/Fm ratios for *D. anglica*, *D. intermedia*, and *D. rotundifolia* according to substrate type (A–D), where: A—water, B—peat, C—*Sphagnum* mosses, D—sand. Different letters (a, b) represent significant differences at *p* < 0.05 probability level according to RIR Tukey’s post hoc test using the Compact Letter Display (CLD) methodology.

**Figure 9 ijms-24-09823-f009:**
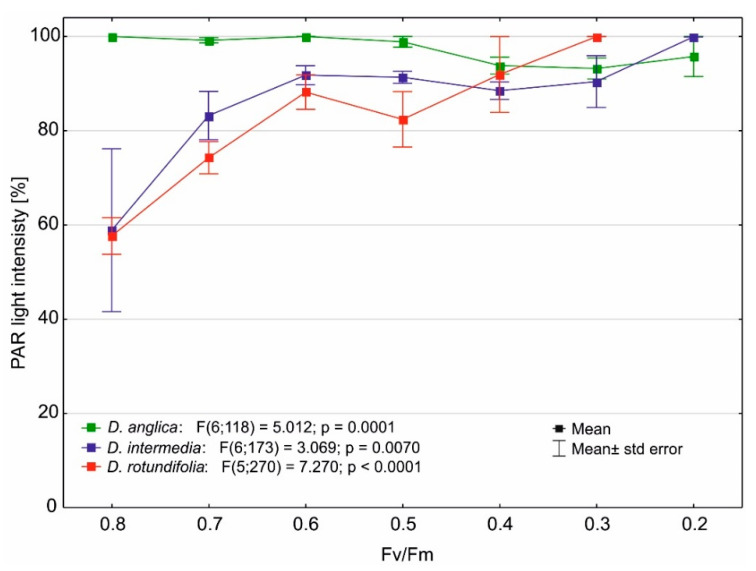
Changes in the value of the Fv/Fm ratio as a function of PAR light intensity.

**Figure 10 ijms-24-09823-f010:**
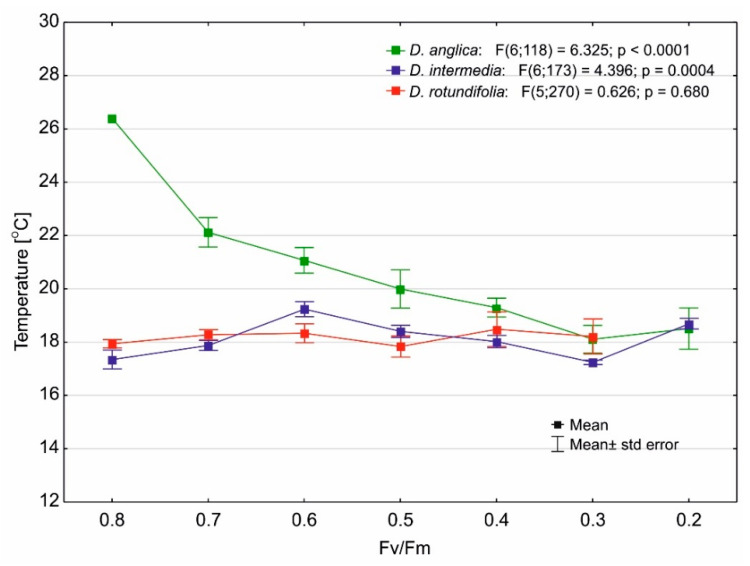
Changes in the maximum quantum yield of photosynthesis as a function of substrate temperature.

**Figure 11 ijms-24-09823-f011:**
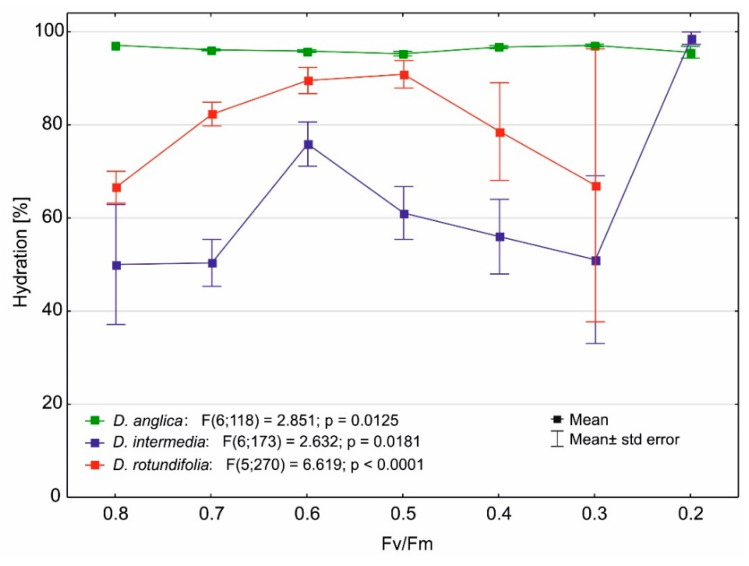
Changes in the maximum quantum yield of photosynthesis as a function of substrate hydration.

**Figure 12 ijms-24-09823-f012:**
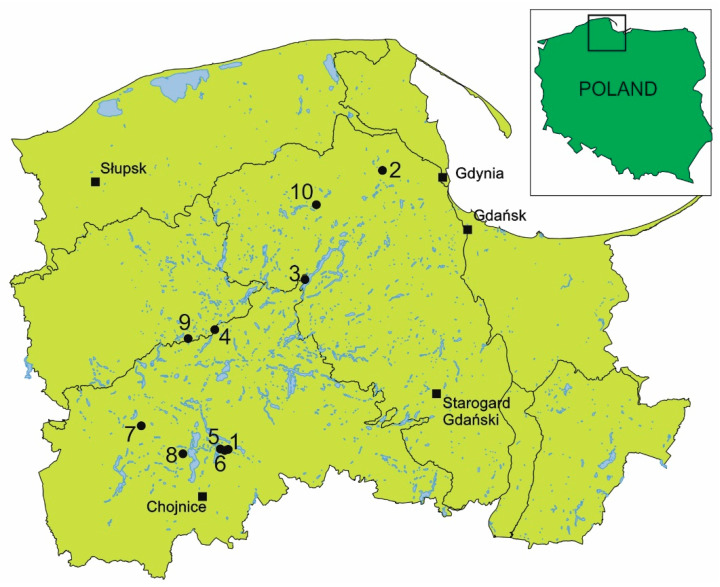
Location of the studied sites (1–10; explanations compare with Table 1).

**Table 1 ijms-24-09823-t001:** Coordinates of study sites (1–10) and occurrences of individual species of the genus *Drosera*.

No.	Coordinates	*D. rotundifolia*				*D. anglica*				*D. intermedia*			
		W	P	M	S	W	P	M	S	W	P	M	S
1.	53°48′52.9″ N 17°38′27.7″ E	+		+		+				+	+		
2.	54°30′31.8″ N 18°16′40.7″ E		+	+		+							
3.	54°14′17.3″ N 17°57′37.0″ E			+			+	+					
4.	54°06′25.8″ N 17°34′55.8″ E	+		+		+	+			+	+		
5.	53°48′58.3″ N 17°36′58.1″ E		+		+								+
6.	53°48′51.2″ N 17°38′00.7″ E				+								+
7.	53°52′03.8″ N 17°16′54.5″ E				+								+
8.	53°48′10.1″ N 17°27′29.8″ E			+									
9.	54°05′11.4″ N 17°28′09.6″ E	+		+		+				+		+	
10.	54°25′27.5″ N 18°00′06.5″ E			+									
No. of plants		18	30	174	54	83	18	24	0	66	12	12	90

Explanations: 1—peatbog near Męcikał (Zaborski Landscape Park), 2—peatbog near Lake Zawiat (ecological area), 3—peatbog near Żuromino, 4—peatbog near Studzienice (projected nature reserve), 5—shore of Lake Sosnówek (PN Bory Tucholskie buffer zone, Zaborski LP), 6—shore of Lake Moczadło (Nature Reserve Lake Moczadło, Zaborski LP), 7—shore of Lake Krasne (Lake Krasne Nature Reserve), 8—peatbog by Lake Małe Łowne (Lake Małe Łowne Nature Reserve), 9—peatbog by Lake Leniwe (Lisia Kępa Nature Reserve), 10—peatbog by Lake Kamienne (Żurawie Błota Nature Reserve). Substrate type: W—water, P—peat, M—sphagnum mosses, S—sand.

**Table 2 ijms-24-09823-t002:** Environmental conditions of sundews.

	*D. anglica*		*D. intermedia*		*D. rotundifolia*	
	Mean	SD	Mean	SD	Mean	SD
PAR light intensity [%]	97.6	6.0	87.9	22.3	72.9	34.9
Temperature [°C]	20.6	3.1	18.4	1.6	18.1	1.97
pH	4.09	0.57	4.78	0.66	4.05	0.67
Conductivity [mS cm^−1^]	64.3	31.3	38.7	27.5	82.7	53.1
Hydration [%]	96.2	1.5	61.6	37.1	78.9	28.3
Organic matter [%]	93.6	2.1	51.6	42.7	75.4	33.4

**Table 3 ijms-24-09823-t003:** Individual architectures of sundews (Mean and Standard Deviation (SD)).

	*D. anglica*		*D. intermedia*		*D. rotundifolia*	
	Mean	SD	Mean	SD	Mean	SD
Rosette width [cm]	8.88	2.92	5.55	1.82	3.26	1.01
Main axis height [cm]	1.33	0.65	0.84	0.69	0.94	0.90
Length of roots [cm]	5.95	2.75	5.95	4.81	3.86	1.80
Number of inflorescences	0.6	0.84	0.7	0.8	0.5	0.7
Number of flowers	2.3	2.87	2.6	2.9	2.0	3.3
Number of live leaves	6.2	3.3	22.7	10.8	7.1	2.6
Number of dead leaves	4.8	1.7	9.4	5.2	5.4	2.0

**Table 4 ijms-24-09823-t004:** Fv/Fm ratio values for *Drosera* species.

Species	N	Mean	SD	Median	Min	Max
*D. anglica*	125	0.543	0.154	0.597	0.228	0.764
*D. intermedia*	180	0.571	0.118	0.572	0.248	0.800
*D. rotundifolia*	275	0.677	0.111	0.706	0.328	0.831
Total	580	0.616	0.137	0.645	0.228	0.831

**Table 5 ijms-24-09823-t005:** Correlations of the Fv/Fm ratios, with environmental traits for individual *Drosera* species. The correlation coefficients marked in bold are significant, with *p* < 0.05.

Species	Environmental Feature					
	pH	Conductivity	Temperature	PAR Light Intensity	Hydration	Organic Matter
*D. anglica*	**−0.259**	−0.152	**0.468**	**0.380**	−0.146	0.146
*D. intermedia*	**−0.171**	−0.115	−0.067	**−0.176**	−0.007	0.027
*D. rotundifolia*	0.071	−0.034	−0.069	**−0.314**	**−0.230**	**−0.267**

**Table 6 ijms-24-09823-t006:** Results of correlation of the Fv/Fm ratio with environmental characteristics on the distinguished substrate types. The correlation coefficients marked in bold are significant, with *p* < 0.050.

	Environmental Traits					
Type of Substrate	pH	Conductivity	Temperature	PAR Light Intensity	Hydration	Organic Matter
*D. anglica*						
water	−0.023	**−0.576**	**0.654**	**0.397**	−0.065	0.140
peat	**−0.552**	**0.675**	**0.533**	0.352	−0.457	**−0.587**
sphagnum	**−0.811**	**0.950**	0.290		**−0.495**	0.155
*D. intermedia*						
water	**−0.751**	**0.599**	**−0.681**	**0.710**	−0.056	−0.143
peat	**−1.0**	**1.0**	**1.0**	**1.0**	**−1.0**	**−1.0**
sphagnum	**1.0**	**1.0**	**−1.0**		**−1.0**	**−1.0**
sand	**−0.573**	0.012	0.069	**−0.290**	0.191	**0.540**
*D. rotundifolia*						
water	0.172	**−0.762**	0.207	0.199	**0.937**	**0.747**
peat	0.222	**0.314**	0.031	**−0.632**	**−0.556**	**−0.398**
sphagnum	0.125	0.103	−0.117	−0.008	−0.146	**0.486**
sand	**−0.590**	0.087	−0.052	**−0.569**	**0.463**	0.185

## Data Availability

The datasets used and/or analyzed during the current study are available from the corresponding author on reasonable request.

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
