# Peer review of "Effects of Environmental Conditions on the Individual Architectures and Photosynthetic Performances of Three Species in Drosera"

_ijms, 2023, doi:10.3390/ijms24129823_

Round 1
Reviewer 1 Report
My comments are as follows,
1. It is suggested to change the title as " Effects of environmental conditions on the individual architectures and photosynthetic performances of three species in Drosera”.
2. In abstract, the authors mentioned that 581 individuals in this study were measured, are all samples of Drosera excluded from hybridization?
3. The introduction should be concise and brief, e.g. is the source of the three species name in this genus is important and relevant to present study? If not, it is suggested to be deleted, including D. rotundifolia, Drosera anglica, Drosera intermedia. The same as the detailed introduction of the status of endangered and protected of the genus.
4. Try to map the 10 different sampling points and show it in a picture.
5. How many morphological features were measured totally? Are some characters related to environmental factors such as leaf size, thickness, and the number of glandular hairs on the leaf surface should be taken into account?
6. The pictures (Figure 8-10) and description of the three species in different habitats are suggested to be move to the Results “2.2 Individual architectures of sundews” rather than in the discussion section.
7. In results, are there significant differences in environmental factors in the habitats of the three species, or within the same species? How about P-values analysis of Environmental conditions of sundews?
8. “D. rotundifolia occupies habitats that are the most diverse, and that are often poorly lighted, with the lowest pH but the highest conductivity. But it is the least variable in terms of individual architecture”, In addition to the number of leaves, stem height and root length, does inflorescences length, number of flowers, are significantly related to the environment?
9. In the methods, it showed that 125 individuals of D. anglica, 180 of D. 578 intermedia, and 276 of D. rotundifolia were measured. What are the morphological differences within each species distributed in different sites? Whether correlations between intraspecific habitat and morphological characteristics have been analyzed?
10. The size of some tables is too large, it should be modified to fit the page size, e.g. Table 1, 5, 6 etc.
11. Note the label in the Figure3 is larger than that of the text, it is suggested to change the format.
12. There are too many references, so it is recommended to simplify them.
13. Family name present in text or references is not italicized.

Reviewer 2 Report
Dear authors,
this manuscript includes is a highly interested study on the 3 important European Drosera species and its morphological, physiological and growing characteristics. Although some details are already known from the literature many parts are still unknown.
Nonetheless, this article is too long and includes too many not really required and necessary text parts, graphs and Figures, which clouds the interesting and important points in this study. I would highly recommend to publish this article after revision for shortening of some text parts – including Figures which are not really needed for the main statements of this study. For example, Fig.2 and 3 maybe transferred to a supplement part. Figures of the typical habitus and morphology are visible in Fig 8,9,10– but additionally, one photograph of all 3 species in their typical environment (which corresponds to the results) would be also favorable for this article.
The discussion part should also be structured with subchapters and additional superscription – - the only structure are thematic headings (environmental stress, water stress, light stress….) without a statement; Also some parts from the discussion should be included in the results part (Fig 11.- 13 as well as Fig 8,9,10 – they are places in the discussion and belong predominantly to the results). Redundant parts in the results and the discussion should be omitted.
Also for the conclusion the performance of the statements and findings of this study should be improved.
Some notes on missing information in the manuscript:
Table 1: why is Drosera rotundifolia and D.anglica marked in bold and D. indermedia not? Please explain.
Table 2: why is D. anglica, Mean and SD displayed bold? Please explain.
Table 3: why is D. anglica, Mean and SD displayed bold? Please explain.
Fig 1: the caption of Fig 1 and the running text are fused together. Figure caption should always be as short as possible;
The numbers for the species-name (Drosera anglica (1), D. intermedia (2), and D. rotundifolia (3)) for the graphs in Figure 1 – are coded with: 1-2-3 and with colours (only 1,2,3 ist mentioned in the subtext) – but what does a,b,c and a.,b,b and also a,b,a at the top of the boxplots means? Please explain! This is also true for some other graphs: Fig 4,6,7 – please explain!
Discussion:
line 282-83: because only under such conditions they do have a significant competitive advantage over non-carnivorous plants [48]. – “they do” instead of “do they”
Reviewer 3 Report
In this study by Krzysztof Banaśf entitled "Effects of environmental conditions on the individual architectures and photosynthetic performances of three sundew species Drosera anglica Huds., D. intermedia Hayne., and D. rotundifolia L." well studied the environmental conditions, individual architectures, and photosynthetic efficiencies of three sundew species: Drosera rotundifolia, D. anglica, and D. 11intermedia, which were found in well-preserved peatlands and sandy lake shores in NW Poland. Morphological traits and chlorophyll a fluorescence (Fv/Fm) were measured in 581 individuals of Drosera. D. anglica under conditions of higher pH, less organic matter, and less well-lit habitats. The values of the Fv/Fm ratio were also reported.
Over all the work is well presented and can be considered for publication.
Author Response
Thank you for reviewing our work
Round 2
Reviewer 1 Report
The revised version of the manuscript has been modified according to the comments.
Author Response
Thank you very much for reviewing our work
Reviewer 2 Report
Dear Authors
There were made a lot of changes and the authors performed the main part of the suggestions!
I make only some notes:
The interesting parts (also the synopsis of the "state of the art") of the Introduction are shortened too much for my taste – but I think this was demanded by another reviewer.
In the discussion section – there is no real “discussion style” from line 379 – 428 please add suitable literature.
There are only a few minor shortcomings that should be corrected:
Line 214: “while in compact Sphagnum mats it...” Sphagnum may be changed to italic ...like „while in compact mats of Sphagnum it.....”
The same is true for line 353-354:
“element of Sphagnum-dominated communities in these habitats”
Best regards
